# Synthesis of core@shell catalysts guided by Tammann temperature

Pei Xiong [1,6], Zhihang Xu [1,6], Tai-Sing Wu [2], Tong Yang[1], Qiong Lei[1], Jiangtong Li[1], Guangchao Li[1], Ming Yang [1], Yun-Liang Soo [3], Robert David Bennett[4], Shu Ping Lau [1], Shik Chi Edman Tsang [5] ✉, Ye Zhu [1] ✉ & Molly Meng-Jung Li [1] ✉

Designing high-performance thermal catalysts with stable catalytic sites is an important challenge. Conventional wisdom holds that strong metal-support interactions can benefit the catalyst performance, but there is a knowledge gap in generalizing this effect across different metals. Here, we have successfully developed a generalizable strong metal-support interaction strategy guided by Tammann temperatures of materials, enabling functional oxide encapsulation of transition metal nanocatalysts. As an illustrative example, $Co@BaAl_2O_4$ core@shell is synthesized and tracked in real-time through in-situ microscopy and spectroscopy, revealing an unconventional strong metal-support interaction encapsulation mechanism. Notably, $Co@BaAl_2O_4$ exhibits exceptional activity relative to previously reported core@shell catalysts, displaying excellent long-term stability during high-temperature chemical reactions and overcoming the durability and reusability limitations of conventional supported catalysts. This pioneering design and widely applicable approach has been validated to guide the encapsulation of various transition metal nanoparticles for environmental tolerance functionalities, offering great potential to advance energy, catalysis, and environmental fields.

Transition metal nanoparticles ($M_T$ NPs) exhibit excellent activities for many important catalytic reactions, such as methane ($CH_4$) or ammonia ($NH_3$) conversions[1–5], yet stability drawbacks hinder their practical applications. This is because the highly active surface atoms are generally thermodynamically unstable, leading to severe NPs sintering via secondary nucleation, recrystallization, ripening, particle migration, and coalescence[6–8]. To preserve the NPs' stability, different approaches have been developed. Among them, the core@shell architecture offers an advantageous structure by isolating the active nanoparticle cores with outer shells to prevent sintering during high-temperature catalytic reactions[9,10]. The most commonly used core@shell synthesis method is to physically confine the NPs in porous materials (e.g., silica, alumina, zeolites, and so forth.)[11,12]. Despite various choices and vast sources of confining materials, the limitation of active site blockage by the inert supports is often observed, thereby diminishing the catalytic performance[13].

Alternatively, strong metal-support interactions (SMSI) have been engineered to produce core@shell structures for stabilizing and modifying active metal NPs. For example, on the important industrial $Cu/ZnO/Al_2O_3$ methanol synthesis catalyst, it is found that the partially reduced zinc oxides can interact with metallic Cu NPs via the SMSI effect[14], forming a protective layer to prevent the sintering of Cu NPs. Due to the unique geometric and sintering-resistant interfacial interactions, various SMSI metal and support compositions have been

[1]Department of Applied Physics, The Hong Kong Polytechnic University, Hong Kong, China. [2]National Synchrotron Radiation Research Center, Hsinchu 30076, Taiwan. [3]Department of Physics, National Tsing Hua University, Hsinchu 30013, Taiwan. [4]CSIRO Energy, Clayton Laboratories, Clayton South, VIC 3168, Australia. [5]Wolfson Catalysis Centre, Department of Chemistry, University of Oxford, Oxford OX1 3QR, UK. [6]These authors contributed equally: Pei Xiong, Zhihang Xu. ✉e-mail: edman.tsang@chem.ox.ac.uk; yezhu@polyu.edu.hk; molly.li@polyu.edu.hk

explored and studied[15]. It is observed that at the metal-support interfaces, the reducible metal oxide substrates (e.g., $TiO_2$, $V_2O_3$, $CeO_2$, and $Ta_2O_5$) can be thermally activated and migrated onto metal NPs surface (typically platinum group metals like Pt, Pd, and so on) under hydrogen ($H_2$) atmosphere at high temperature, forming complete encapsulation of the NPs by the oxides[15–17]. These SMSI metal@support configurations not only dramatically improve the thermodynamic stability of the NPs against sintering but also impart tunable versatility of the catalytic properties via modulating the core@shell interface[18–20].

Unfortunately, the high temperature required in the thermal activation step of SMSI restricts the application of this strategy to NPs with low melting points. For those with inherently low melting temperatures, such as Fe, Co, Ni, Cu, Au, and so on, the NPs tend to sinter before the encapsulation occurs[21]. Hence, to achieve successful encapsulation of $M_T$ NPs through SMSI, it is critical to activate the supporting metal oxides with sufficient mobility before the $M_T$ NPs sinter. Several successes have been reported, such as $HCO_x$-adsorbate-mediated SMSI encapsulation on $TiO_2$- and $Nb_2O_5$-supported Rh NPs[22], $CO_2$-enhanced redox-inert MgO encapsulation on Au NPs[23], and the adsorbate-induced SMSI reconstruction of the commercial Cu/ZnO/$Al_2O_3$ that maximizes the activity and stability[24]. However, these methods are highly material-specific and thus hard to be extended to other systems. Therefore, novel generalizable core@shell synthesis strategies are highly demanded.

In this work, we establish a general and straightforward approach for achieving thermal-induced SMSI encapsulation on $M_T$ NPs. The approach involves utilizing the empirical Tammann temperature ($T_{Tam}$) phenomenon, which describes the atoms in the solid-state

materials becoming loosened with higher mobility and reactivity at temperatures higher than c.a. 1/2 of the melting point[25]. We leverage the advantage of the low $T_{Tam}$ compounds that can be thermally activated and reacted at relatively low temperatures. Once a coating of low $T_{Tam}$ compounds is formed on $M_T$ NPs, further overlayer functionalization can be realized through a solid-state reaction, resulting in the formation of the highly thermally stable protective layer of the catalyst support.

## Results and Discussion

### Synthesis rationale for thermal encapsulation based on $T_{Tam}$

Herein, we synthesize $M_T$ NPs by in-situ reduction from the corresponding $M_T$ oxides ($M_TO_x$) to avoid the troublesome multi-step processes and handling of active $M_T$ NPs materials. To establish SMSI encapsulation, we refer to the $T_{Tam}$ as an indicator to select suitable encapsulating materials, i.e., supports, for the targeted metal catalysts. More specifically, as shown in Fig. 1a, the $T_{Tam}$ of the encapsulating materials, denoted as $T_{Tam}$(support), should be lower than the reduction temperature ($T_{red}$) of $M_TO_x$, denoted as $T_{red}(M_TO_x)$. According to this rule, the support can facilitate more efficient diffusion, acting as a mobile media to establish contact with the $M_TO_x$ at $T_{Tam}$(support). During the formation of $M_T$ NP at $T_{red}(M_TO_x)$, encapsulation can take place due to the tendency to reduce the surface energy of $M_T$ NPs. This process helps prevent the excessive growth of $M_T$ particles. Meanwhile, the decomposition temperature ($T_{dec}$) of the encapsulating materials, denoted as $T_{dec}$(support), must be higher than the $T_{red}(M_TO_x)$ to avoid the loss of high mobility due to the phase/structure change. Based on these considerations, if one support

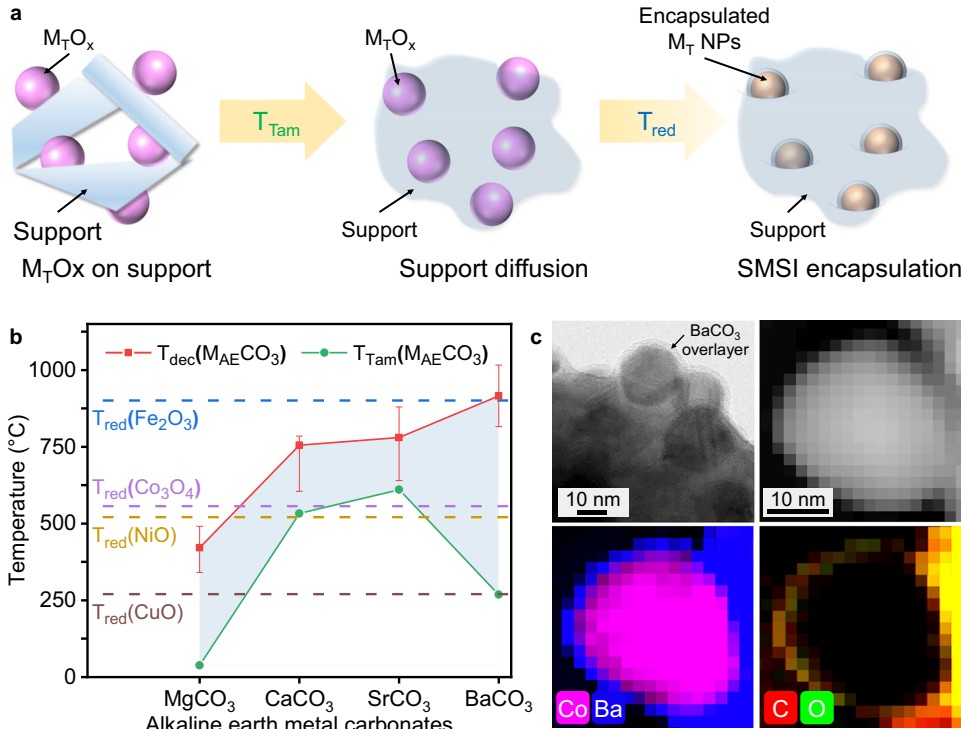

**Fig. 1 | Synthesis rationale for thermal encapsulation based on Tammann temperature ($T_{Tam}$). a** Schematic illustration of the designed synthesis rationale for this thermal-induced encapsulation strategy guided by Tammann temperature; $M_TO_x$ indicates transition metal nanoparticles oxides, and $M_T$ NPs indicates transition metal nanoparticles. **b** Schematic of the estimated suitable temperature range (or encapsulable window, shadowed in blue) to enable encapsulation on in-situ formed $M_T$ NPs with alkaline earth metal carbonates ($M_{AE}CO_3$) through thermal-induced strong metal-support interaction (SMSI); $T_{Tam}(M_{AE}CO_3)$ indicates Tammann temperature of $M_{AE}CO_3$, $T_{dec}(M_{AE}CO_3)$ indicates the decomposition

temperature of $M_{AE}CO_3$, and $T_{red}$ indicates the reduction temperature of $M_TO_x$; the upper and lower limits of the error bars for the decomposition temperature are taken respectively from the starting and ending temperatures of the decomposition peak of the substance. **c** Transmission electron microscopy (TEM), scanning transmission electron microscopy (STEM) images and electron energy loss spectroscopy (EELS) element distribution maps of Co@$BaCO_3$ (additional Co@$BaCO_3$ EELS elemental mapping analysis can be found in Supplementary Fig. 8). Source data are provided as a Source Data file.

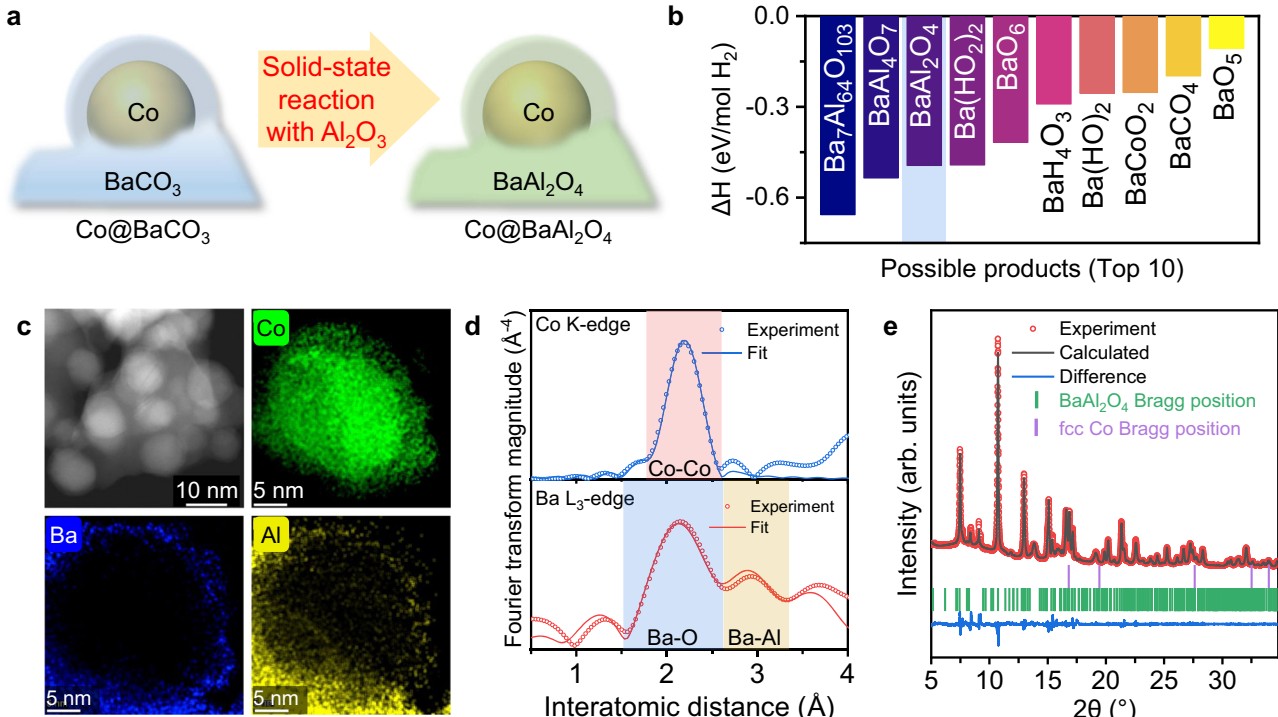

**Fig. 2 | One-step metal encapsulation and overlayer functionalization.**
**a** Schematic illustration of the subsequent overlayer functionalization step following the thermal-induced strong metal-support interaction (SMSI) encapsulation. **b** Rapid computational screening of reaction enthalpy changes ($\Delta H$) for forming possible products from solid-state reactions between $BaCO_3$, $Al_2O_3$ and $Co_3O_4$ in $H_2$. $\Delta H$ were obtained from the Materials Project database. The lower the $\Delta H$, the more stable the reaction product. **c** Scanning transmission electron microscopy (STEM) image and energy dispersive X-ray spectroscopy (EDS) element distribution maps of $Co@BaAl_2O_4$ (More details in Supplementary Fig. 11d). **d** Extended X-ray absorption fine structure (EXAFS) fitting curves in R-space for the Co K-edge and Ba $L_3$-edge of $Co@BaAl_2O_4$. **e** Rietveld refinement of the synchrotron X-ray diffraction (SXRD) pattern ($\lambda = 0.5904$ Å) of $Co@BaAl_2O_4$. Source data are provided as a Source Data file.

material can fit in the simple rule of $T_{Tam}(support) < T_{red}(M_TO_x) < T_{dec}(support)$, then it is expected to exhibit high mobility to achieve SMSI encapsulation on $M_T$ NPs at $T_{red}(M_TO_x)$.

As the proof-of-concept demonstration, we choose low $T_{Tam}$ alkaline earth metal (common catalyst promoters) in the carbonate form ($M_{AE}CO_3$) as the candidate support. The phase-changing temperatures of the common non-noble $M_TO_x$ ($M_T$ = Fe, Co, Ni, Cu) and $M_{AE}CO_3$ ($M_{AE}$ = Mg, Ca, Sr, Ba) combinations are mapped in Fig. 1b. The $T_{dec}(M_{AE}CO_3)$ and $T_{red}(M_TO_x)$ are measured by $H_2$ temperature-programmed decomposition and reduction experiments, respectively (see details in Supplementary Figs. 6 and 7, Supplementary Table 1), while the $T_{Tam}(M_{AE}CO_3)$ are retrieved from literature[21,26–28]. In Fig. 1b, the region sandwiched by $T_{dec}(M_{AE}CO_3)$ and $T_{Tam}(M_{AE}CO_3)$ (shadowed in blue) indicates the suitable temperature window (or encapsulable window) to theoretically allow for the facile SMSI encapsulation on the $M_T$ NPs if the corresponding $T_{red}(M_TO_x)$ falls into this range. A test synthesis on the Co and $BaCO_3$ combination has been attempted to prove our concept. We obtain uniformly distributed Co NPs encapsulated in $BaCO_3$ (i.e., $Co@BaCO_3$) through one-step thermal treatment at the temperature around $T_{red}(Co_3O_4)$ in $H_2$ atmosphere (see details in Fig. 1c, Supplementary Fig. 8 and Supplementary Table 3). This result demonstrates the viability of our proposed strategy for the convenient thermal encapsulation of $M_T$ NPs guided by Tammann temperatures.

**One-step metal encapsulation and overlayer functionalization**
To take full advantage of the high reactivity of the low-$T_{Tam}$ $M_{AE}CO_3$, we propose further functionalizing the $BaCO_3$ overlayer during the Co NP formation. Considering alkaline earth metal aluminates (i.e., $(M_{AE})_xAl_yO_z$) are classic catalyst support materials with high thermal resistance, good hydrothermal stability, and prominent interactions

with the active species[29,30], we attempt to introduce $Al_2O_3$ into the encapsulation system to engineer the overlayer functionalization (Fig. 2a). By employing rapid computational sorting of the formation energies of different aluminate products from the solid-state reaction between $BaCO_3$ and $Al_2O_3$ (Fig. 2b and Supplementary Fig. 9), we identify that the most suitable overlayer aluminate composition is $BaAl_2O_4$, and reject the more stable $BaAl_4O_7$ and $Ba_7Al_{64}O_{103}$ because they require harsh synthesis conditions (>1100 °C)[31,32].

To realize Co NP formation, $BaCO_3$ encapsulation, and $BaAl_2O_4$ formation in a single thermal treatment, we start with mixing the Co, Ba and Al metal precursors with a composition of 40Co:20Ba:40Al through an automatically pH-controlled co-precipitation method, followed by temperature- and flow-controlled air-calcination (Supplementary Fig. 1). After the thermal treatment in $H_2$, the formation of Co NPs and $BaAl_2O_4$ can be confirmed by the X-ray diffraction (XRD) patterns (Supplementary Fig. 10a). Scanning transmission electron microscopy (STEM) and energy dispersive X-ray spectroscopy (EDS) mapping (Fig. 2c and Supplementary Fig. 11d) clearly show the encapsulation structure with the well-rounded $BaAl_2O_4$ overlayer in close contact with the Co NPs. The encapsulated Co NPs show relatively uniform size distributions in the range of 10-15 nm, in line with the crystal size derived from XRD (Supplementary Fig. 10b and Supplementary Table 4). Transmission electron microscopy (TEM) images and corresponding fast Fourier transform (FFT) patterns of $Co@BaAl_2O_4$ reveal the fcc Co and $BaAl_2O_4$ lattice fringes (Supplementary Fig. 11), which is fully consistent with the Co K-edge and Ba $L_3$-edge extended X-ray absorption fine structure (EXAFS) (Fig. 2d, Supplementary Figs. 12 and 13, and Supplementary Tables 5 to 8) and the synchrotron X-ray diffraction (SXRD) results (Fig. 2e).

Interestingly, noticeable oxygen vacancies in the $BaAl_2O_4$ overlayer of $Co@BaAl_2O_4$ can be detected by X-ray photoelectron

spectroscopy (XPS) and electron paramagnetic resonance (EPR) spectroscopy (Supplementary Fig. 14a, b). In addition, Ar adsorption-desorption analyses reveal the porous structure of Co@BaAl$_2$O$_4$ (Supplementary Fig. 14c, d), presumably due to the presence of oxygen vacancies when synthesizing BaAl$_2$O$_4$ under H$_2$ environment. This porous feature is particularly important for catalysis as they impart the good permeability of core@shell structure for the reactant molecules to access the active metal core while maintaining high stability from encapsulation.

It is important to note that the observed Co@BaAl$_2$O$_4$ encapsulation cannot be realized by direct synthesis using BaAl$_2$O$_4$ (see details in Supplementary Fig. 15 and Supplementary Table 9), reflecting the necessity of employing low T$_{Tam}$ BaCO$_3$ for our proposed SMSI encapsulation process. In addition, severe Fe or Cu NP coalescence is observed in the control experiments involving M$_T$O$_x$-M$_{AE}$CO$_3$ combinations where the T$_{red}$(M$_T$O$_x$) falls either outside or on the edge of the encapsulable window, further validating our proposed thermal encapsulation and sintering prevention strategy guided by Tammann temperature (Supplementary Figs. 16 and 17).

## Real-time observation for the encapsulation mechanism

The Co@BaAl$_2$O$_4$ synthesis process is further investigated to explore the encapsulation formation pathway and mechanism. Starting with the 40Co:20Ba:40Al precursor, the elemental composition analyses indicate a similar atomic ratio of Co, Ba, and Al at c.a. 2:1:2 at different stages of the synthesis process (i.e., as-precipitated, as-calcined, and thermally treated in H$_2$, Supplementary Fig. 19 and Supplementary Table 11). Yet distinct morphologies are revealed by TEM, where the encapsulation is only formed after the thermal treatment in H$_2$ (Supplementary Fig. 20). Direct real-time visualization of the encapsulation process during the H$_2$ thermal treatment is then conducted using in-situ atmospheric STEM equipped with an environmental cell (see Supplementary Fig. 4 for experiment setup and details). At 25 °C, the calcined precursor is initially a mixture of plate-like Co$_3$O$_4$ and rod-like Al$_2$O$_3$ NPs in contact with much larger bulk of BaCO$_3$ (Supplementary Fig. 21). STEM image indicates a clear boundary between the dark Co$_3$O$_4$-Al$_2$O$_3$ mixture and the bright BaCO$_3$ bulk particle (Fig. 3a). When the temperature is ramped to above 400 °C (Fig. 3b and Supplementary Movie 1), the boundary of the BaCO$_3$ particle blurs and some darker pores (marked by dashed circles) appear in the interior, which can be attributed to the increased lattice mobility and subsequent outward diffusion as evidenced by the boundary propagation (the dotted lines). Raising the temperature to 500 °C leads to more obvious diffusion of BaCO$_3$ with dark pores growing inside, agreeing well with the expanding Ba element distribution observed from EDS elemental mapping (Supplementary Fig. 22a, b). In contrast, no obvious morphology change is observed in the Co/Al-containing area until the bright spheres emerge at 600 °C and become more identifiable at 700 °C (Fig. 3c and Supplementary Movie 2), indicating the in-situ formation of Co NPs from reducing Co$_3$O$_4$. The Co NPs are surrounded by highly mobile or liquified BaCO$_3$, forming the initial encapsulation. When the cell is heated from 700 to 800 °C, a dynamic reaction can be seen with BaCO$_3$ decomposition at the expense of the Al$_2$O$_3$ rods in proximity, implying the solid-state formation of the functional BaAl$_2$O$_4$ overlayer (Fig. 3d, Supplementary Fig. 22c and Supplementary Movie 3), as confirmed by selected area electron diffraction (SAED) (Supplementary Fig. 22d). The observed morphology evolution is summarized in Fig. 3e, in full accordance with our proposed SMSI-induced encapsulation strategy guided by Tammann temperature.

XRD and in-situ Raman spectroscopy equipped with online MS (see details in Supplementary Fig. 3) are also employed to supplement the real-time phase evolution and the solid-state reactions. At the initial stage, typical XRD patterns of BaCO$_3$ and Co$_3$O$_4$ are recorded (Fig. 3f), with the absence of Al$_2$O$_3$ fingerprint peaks, presumably overwhelmed by the signals of Co$_3$O$_4$ or the nano crystallinity of

gamma phase Al$_2$O$_3$ (Supplementary Fig. 21d). As the temperature of thermal treatment increases to 400 °C, the characteristic peaks of Co$_3$O$_4$ at 31.27° and 36.84° fade with the simultaneous appearance of two new peaks at 44.22° and 51.52°, which can be assigned to the fcc Co. At 700 °C, the characteristic diffraction signal of BaCO$_3$ is replaced by that of BaAl$_2$O$_4$, indicating the completion of the solid-state reaction between BaCO$_3$ and Al$_2$O$_3$. Similarly, in-situ Raman spectroscopy reveals the gradual disappearance of Co$_3$O$_4$ fingerprints around ~490, ~525, ~610, and ~690 cm$^{-1}$ before 400 °C[33], and the replacement of BaCO$_3$ fingerprints[34] around ~139, ~698, and ~1060 cm$^{-1}$ by those of BaAl$_2$O$_4$ around ~193, ~271, ~491, ~581, and ~680 cm$^{-1}$ at 700 °C (Fig. 3g)[35]. Online MS is used to monitor the outlet gas during the thermal treatment under H$_2$ flow. It can be observed that Co$_3$O$_4$ is reduced during 300-500 °C, with H$_2$ consumed and H$_2$O released. As the temperature is further increased to 700 °C, both CO and H$_2$O peaks rise rapidly while consuming H$_2$, which is consistent with the solid-state reaction between BaCO$_3$ and Al$_2$O$_3$. In other words, these results echo the observed phase change process in the in-situ STEM. Note that the discrepancies in some phase-changing temperatures recorded by different characterization techniques are due to the distinct in-situ conditions, which do not affect the trend of the identified phase evolution and encapsulation mechanism.

All the evidence collectively indicates that (i) low T$_{Tam}$ BaCO$_3$ exhibits sufficient mobility and can migrate onto the in-situ formed Co NPs, and (ii) BaCO$_3$ could readily react with Al$_2$O$_3$ to form BaAl$_2$O$_4$ as the final encapsulation overlayer (i.e., BaCO$_3$ + Al$_2$O$_3$ + H$_2$ → BaAl$_2$O$_4$ + CO + H$_2$O), thanks to the high reactivity of low-T$_{Tam}$ BaCO$_3$ in the H$_2$ atmosphere at a higher temperature. As a result, the complete encapsulation is realized on the Co NP surface with BaAl$_2$O$_4$ by forming a well-defined core@shell configuration.

## Excellent catalytic performance of encapsulated nanocatalyst

To assess the impact of BaAl$_2$O$_4$ encapsulation on the catalytic performance of the Co NPs at high temperatures, a series of Co nanocatalysts, both with and without encapsulation, are evaluated for their performance in thermal catalytic NH$_3$ decomposition reaction. Typically, this reaction requires a temperature of approximately 650-800 °C for non-noble metal catalysts to achieve an NH$_3$ conversion of over 80% without sacrificing the weight hourly space velocity (WHSV)[36,37]. Surprisingly, the Co@BaAl$_2$O$_4$ achieves almost complete conversion (~99%) at a much lower temperature of only 500 °C, at a high WHSV of 30,000 mL g$_{cat}^{-1}$ h$^{-1}$ (Supplementary Fig. 25 and Supplementary Table 12), obtaining the H$_2$ production rate of 30.9 mmol H$_2$ g$_{cat}^{-1}$ min$^{-1}$ and making it the best core@shell catalysts ever reported for NH$_3$ decomposition (Fig. 4a). Compared to the state-of-the-art NH$_3$ decomposition catalysts, the encapsulated Co@BaAl$_2$O$_4$ also represents one of the most effective catalysts in both non-noble and noble catalysts records (Supplementary Fig. 26 and Supplementary Table 13).

The most significant benefit of the core@shell catalyst configuration is demonstrated in long-term performance evaluation. After 50 hours of time-on-stream at 550 °C, the H$_2$ production of the non-encapsulated catalysts, Co NPs supported on Al$_2$O$_3$ (Co/Al$_2$O$_3$) and on BaAl$_2$O$_4$ (synthesis details see Supplementary Fig. 15), significantly decreases, indicating poor long-term activity due to the severe sintering of Co particles in high-temperature reaction condition (Fig. 4b, c, Supplementary Figs. 27 and 28, and Supplementary Tables 14 and 15). In sharp contrast, the Co@BaAl$_2$O$_4$ nanocatalyst maintains a constantly high H$_2$ production rate over 600 hours under identical testing condition (Fig. 4b). No significant change in morphology is observed for the post-reaction Co@BaAl$_2$O$_4$ (Fig. 4d, Supplementary Fig. 29, and Supplementary Table 16), indicating its superior thermochemical stability due to the advantageous core@shell structure. The EXAFS of the post-reaction Co@BaAl$_2$O$_4$ reveals the distinct Co and BaAl$_2$O$_4$ phases (Supplementary Fig. 29d,

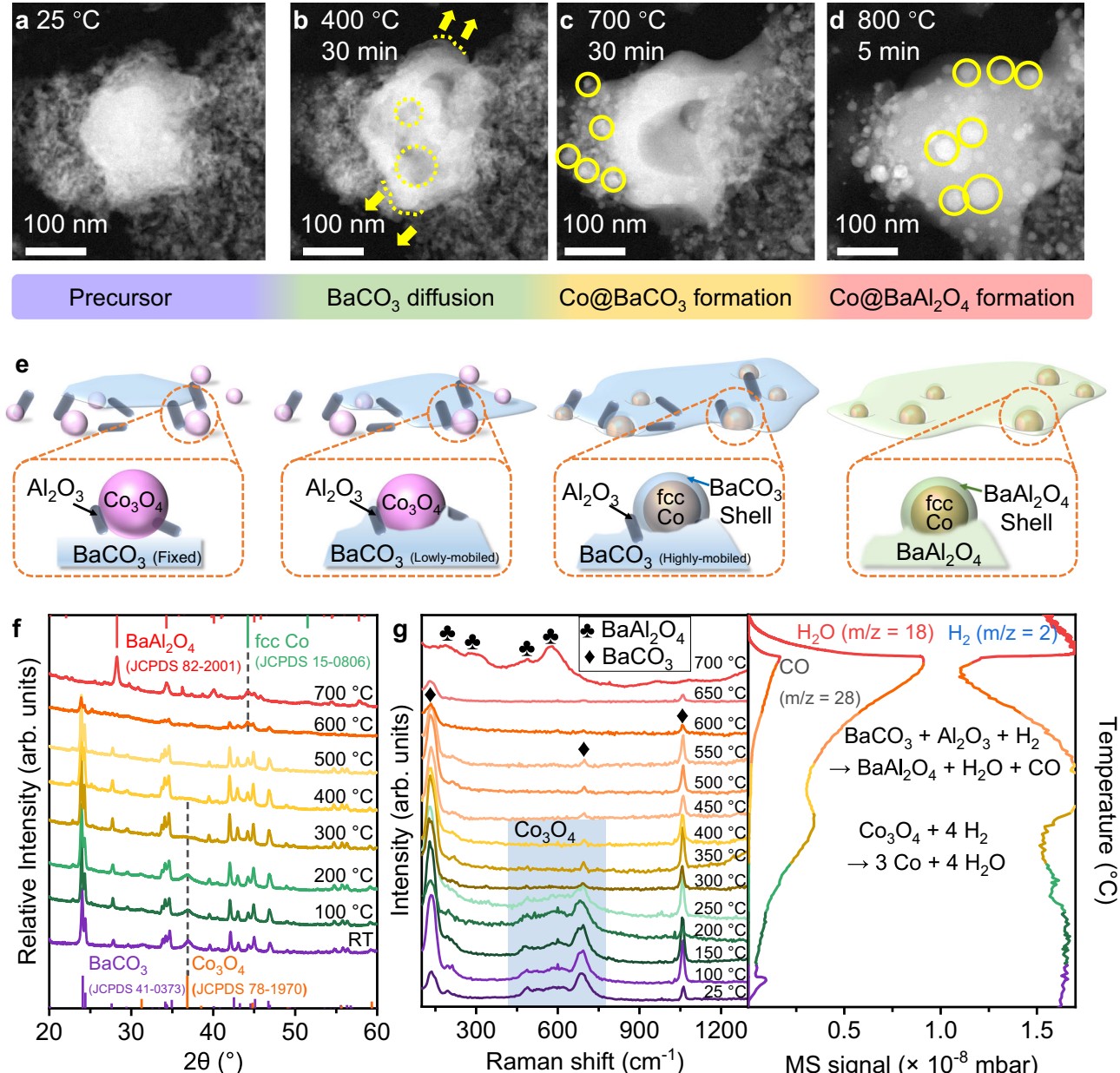

**Fig. 3 | Real-time observation for the encapsulation mechanism.** In-situ scanning transmission electron microscopy (STEM) images at the initial state at room temperature (**a**), BaCO$_3$ diffusion stage when the temperature is above T$_{Tam}$(BaCO$_3$) (**b**) (the yellow dotted lines represent the expansion of the BaCO$_3$ boundary, and the dashed yellow circles show the vacancies inside the BaCO$_3$), Co nanoparticle (NP) formation stage when the temperature is above the reduction temperature (T$_{red}$) of Co$_3$O$_4$ (**c**) (the yellow circles indicate the Co nucleation and growth), and Co@BaAl$_2$O$_4$ formation stage after the solid-state reaction between BaCO$_3$ and Al$_2$O$_3$ (**d**). **e** Schematic illustration of the mechanism for this thermal-induced encapsulation strategy guided by Tammann temperature. **f** X-ray diffraction (XRD) patterns of the 40Co:20Ba:40Al reduced at different temperatures ranging from room temperature to 700 °C. **g** In-situ Raman spectra of the 40Co:20Ba:40Al reduced at different temperatures ranging from room temperature to 700 °C, and mass spectrometry (MS) signals of the outlet gas monitored during the H$_2$ reduction process (tracking m/z = 2 for H$_2$, m/z = 18 for H$_2$O and m/z = 28 for CO). Source data are provided as a Source Data file.

Supplementary Tables 17 and 18), suggesting no migration of Co species at the metal-support interfaces during the long-term testing. Remarkably, over a five-month-long experiment, the activity of Co@BaAl$_2$O$_4$ does not decrease despite intentional temperature fluctuations, system ONs/OFFs, and catalyst offload/reload events (Supplementary Fig. 30), demonstrating its excellent environmental tolerance and reusability for practical applications.

Furthermore, Co@BaAl$_2$O$_4$ demonstrates promising outcomes in the CH$_4$ dry-reforming reaction, which typically requires even higher temperatures (>750 °C)[3]. As shown in Supplementary Fig. 31, Co@BaAl$_2$O$_4$ achieves the CH$_4$ and CO$_2$ conversion rates of 81.8% and

90.1%, respectively, nearing the thermodynamic equilibrium value and sustaining the performance for over 60 hours. In comparison to the benchmark catalysts reported in literature (Supplementary Table 19), the high efficiency and excellent stability identify Co@BaAl$_2$O$_4$ as a promising catalyst for the CH$_4$ dry-reforming reaction. These results jointly manifest that our unconventional encapsulation strategy not only effectively stabilizes the nanocatalyst against the common sintering issue but also provides further opportunities to modify the performance for good catalytic applicability. Currently, the origin of the exceptional performance of Co@BaAl$_2$O$_4$ in the above-mentioned thermal catalytic reactions is under careful investigation, with a focus

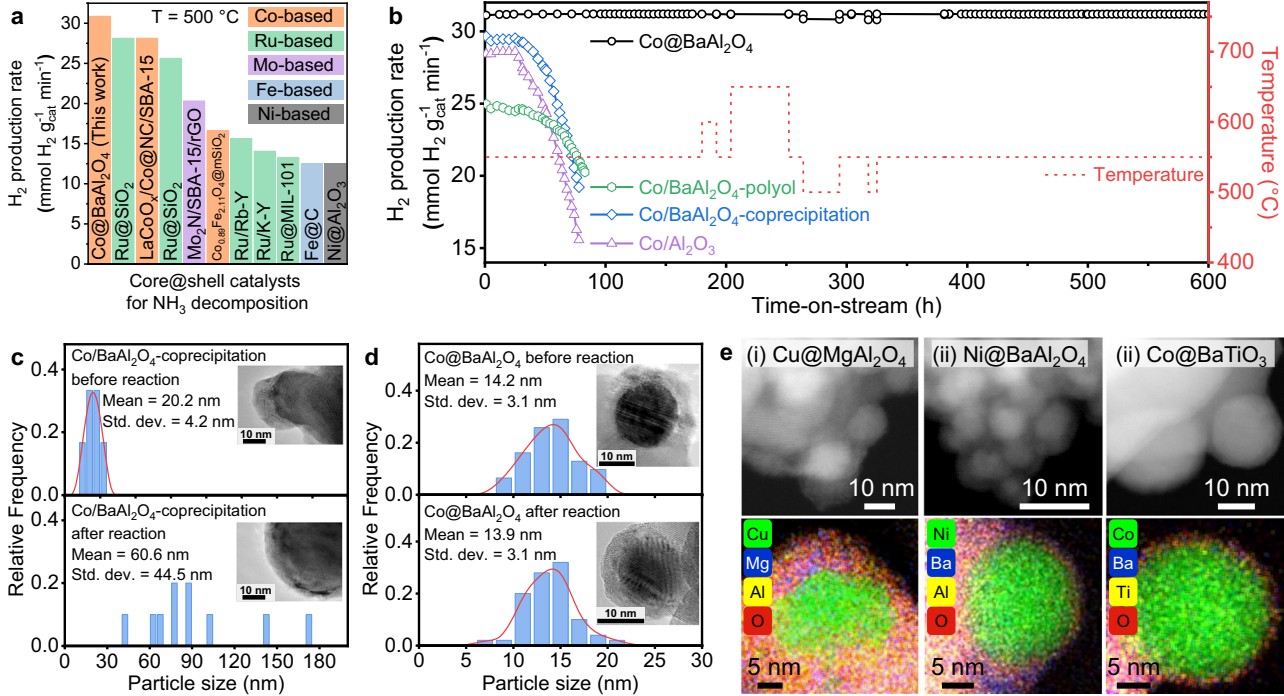

**Fig. 4 | Excellent catalytic performance of encapsulated nanocatalyst.**
**a** Comparison of $H_2$ production rate at 500 °C between the Co@BaAl$_2$O$_4$ and the selected core@shell catalysts used for $NH_3$ decomposition reaction (details see Supplementary Table 11). **b** $H_2$ production rate according to time-on-stream during the $NH_3$ decomposition reaction over the core@shell Co@BaAl$_2$O$_4$ (grey blank balls), and the non-encapsulated catalysts including Co/Al$_2$O$_3$ (blue solid balls), and Co/BaAl$_2$O$_4$ samples prepared by co-precipitation (purple solid balls) and polyol method (green solid balls). The performance was evaluated at weight hourly space velocity (WHSV) of 30,000 mL g$_{cat}^{-1}$ h$^{-1}$. **c** Co NP size distribution histograms of the non-encapsulated Co/BaAl$_2$O$_4$-coprecipitation before and after 80-hour $NH_3$ decomposition reaction, inset shows the transmission electron microscopy (TEM) images of corresponding samples. **d** Co NP size distribution histograms of core@shell Co@BaAl$_2$O$_4$ before and after 80-hour $NH_3$ decomposition reaction, inset shows the TEM images of corresponding samples. **e** scanning transmission electron microscopy (STEM) images and energy dispersive X-ray spectroscopy (EDS) element distribution maps of Cu@MgAl$_2$O$_4$, Ni@BaAl$_2$O$_4$, and Co@BaTiO$_3$. Source data are provided as a Source Data file.

on the notable synergistic effects of metal–support interfaces in the core@shell structure.

## The broad applicability of the core@shell synthesis platform

We have successfully demonstrated that low-$T_{Tam}$ compounds can facilitate thermal-induced SMSI core@shell construction, and subsequent solid-state reactions for overlayer functionalization enable thermal catalytic applicability. Furthermore, to showcase the general applicability of our proposed approach, we also test other $M_T$ and $M_{AE}CO_3$ combinations in the encapsulable window (shadowed in blue in Fig. 1b). The results have validated the effectiveness of our thermal encapsulation and sintering prevention strategy guided by Tammann temperature (Fig. 4e, Supplementary Figs. 18, 32 and 33, and Supplementary Table 20). More convincingly, the replacement of Al$_2$O$_3$ with TiO$_2$ for the overlayer solid-state reaction enables us to achieve BaTiO$_3$ encapsulation on Co NPs (Fig. 4e, Supplementary Fig. 34, and Supplementary Table 20), demonstrating the broad applicability of our strategy across various material matrices (Supplementary Fig. 35).

All these results demonstrate the potential of the Tammann temperature-guided SMSI strategy as a design and synthesis platform for the facile encapsulation of $M_T$ NPs. The tunability of both $M_T$ core and functional oxide overlayer shows promise for fine-tuning the performance of encapsulated nanocatalysts. It is anticipated that other low-$T_{Tam}$ metal compounds, such as chloride, sulfate, nitrate, and many more, can also be applied to construct size-controlled encapsulation structures for different transition metal NPs, either from in-situ reduction or pre-made metal NPs. Computational calculations can assist with rapid composition selections from a broad material matrix for overlayer solid-state reactions, thus unlocking a wide range of possibilities for metal oxides (or non-oxides) to act as NP protectors/

modifiers. Overall, this generalizable thermal encapsulation methodology enables future explorations and predictions for rational and dynamic tuning of nanocatalysts to meet diverse requirements in broad energy and environmental applications.

## Methods

### Synthesis and thermal treatment of catalyst precursors

The precursor mixtures, $mM_T$:$nM_{AE}$:$2nAl$, were synthesised by an automatic pH-controlled co-precipitation method. For a typical preparation procedure, an aqueous solution (50 mL) containing $M_T$, $M_{AE}$ and Al cations with a target molar ratio was prepared by dissolving the corresponding metal nitrates ($M_T$ = transition metal; $M_{AE}$ = alkaline earth metal) in deionised (DI) water. At room temperature, the mixmetal solution was added dropwise into a stirred tank reactor (capacity of 500 mL-2 L) with a 0.5 M Na$_2$CO$_3$ (100 mL) solution under feeding rates of 0.1-2.0 mL min$^{-1}$ regulated by a syringe pump. The mixture was stirred vigorously to ensure efficient mixing. At the same time, the pH of the precipitating solution was carefully maintained at a constant by the dropwise addition of a 4.0 M NaOH solution using another syringe pump. The pH value of the solution should be controlled carefully to form $M_{AE}CO_3$ rather than $M_{AE}(OH)_2$ precipitates. This research used a pH of 12.5 for obtaining BaCO$_3$ and a pH of 9.0 for MgCO$_3$. Once all the pre-measured solutions were added to the tank, the liquid was aged for 16 hours, and then the mixture was filtered and washed with DI water until the pH was close to 7.0. The obtained wet cake solid sample was re-dispersed in 200 mL acetone and stirred at room temperature for 2 hours. Following that, the resultant solid was vacuum filtered, washed thoroughly with acetone and dried overnight in a vacuum oven at room temperature. The precursors are named by their nominal mixed-metal ratios, denoted as $mM_T$:$nM_{AE}$:$2nAl$, where m and n

represent the nominal molar ratios of $M_T$ and $M_{AE}$ in the co-precipitation process, respectively. After getting the $mM_T:nM_{AE}:2nAl$ precipitate, 100 mg of the samples were calcined in the air (denoted as C(temp.)-$mM_T:nM_{AE}:2nAl$) and thermally treated in $H_2/Ar$ (5/95, v/v) at specified temperatures to finally form the possible encapsulation structure, obtaining above 50 mg products denoted as C(temp.)-R(temp.)-$mM_T:nM_{AE}:2nAl$. C(temp.) and R(temp.) indicate the temperatures of air-calcination and $H_2$ treatment processes, and the highest temperature (°C) reached in each step is listed in the brackets. Note: For convenience, those samples which have been observed to be successful are denoted as $M_T@M_{AE}Al_2O_4$. Supplementary Fig. 1 is the schematic representation of the steps involved in this synthesis method.

## Material characterizations

Phases and crystallographic structures of the mentioned samples were characterised by synchrotron X-ray diffraction (SXRD), which were collected using the Powder Diffraction (PD) beamline at the Australian Synchrotron (AS), with a photon energy of 21.0005 keV ($\lambda = 0.5904$ Å). The microstructure and phase information of the samples were characterised by scanning transmission electron microscopy (STEM) in high-angle annular dark field (HAADF) mode and transmission electron microscopy (TEM), using double-Cs-corrected STEM (Spectra 300, TFS, USA) equipped with Super-X energy-dispersive X-ray spectroscopy (EDX) and STEM/TEM (JEM-2100F, JEOL, Japan) combined with a Gatan Enfina electron spectrometer (USA). X-ray absorption fine structure (XAFS) measurements were performed in fluorescence mode using a Lytel detector at beamline BL01C of Taiwan Light Source, National Synchrotron Radiation Research Center (NSRRC). A Si(111) double crystal monochromator (DCM) was used to scan the photon energy. Direct visualisation of the encapsulation process of $Co@BaAl_2O_4$ was achieved using the in-situ atmospheric STEM study (JEOL JEM-2100F). During the in-situ STEM observation, a Protochips atmosphere gas holder was applied, in which the flowing gas can be injected, and the temperature can be controlled. In this research, 380 Torr $H_2/N_2$ (5/5, v/v) with 0.1 mL/min flow rate were used, and the temperature was set from 25 to 800 °C. In addition, the structure evolution during the $H_2$ thermal treatment process was observed by in-situ Raman. About 5 mg samples were loaded into the in-situ Raman cell, with $H_2/Ar$ (5/95, v/v) passing through the samples. A series of Raman spectra were obtained at different temperatures, from room temperature to 700 °C with an interval of 50 °C. Each temperature was set dwelling for 40 min, and the spectra were collected at the 30th minute to get a steady state. In-situ Raman spectra were recorded on a WITEC confocal microscopy system with a laser diode at 532 nm. A 50× objective lens was used to focus the laser on the sample, and the laser spot size was 1 μm.

## NH₃ decomposition performance testing method and setup

To measure the catalytic $NH_3$ decomposition performance of designed catalysts, inside a quartz tube (diameter of 4.5 mm), c.a. 50 mg sieved (45–80 mesh) sample was sandwiched between two layers of quartz wool with a thermocouple placed in contact with the sample. Then high-purity $NH_3$ (≥99.99%) was passed through the catalyst bed with the flow rate controlled by a mass flow controller. The weight hourly space velocity (WHSV) was set as 30,000 mL $g_{cat}^{-1}$ $h^{-1}$ at atmospheric pressure. The concentrations of outlet $N_2$, $H_2$ and $NH_3$ after the reaction were measured online by MS (HPR-20 EGA, Hiden), which was equipped with a quadrupole probe and a secondary electron multiplier detector (850 eV). The accuracy of product analyses was further verified by back titration (Supplementary Fig. 24b). The measured temperatures range from 450 to 650 °C with 50 °C as an interval, and a steady state was reached by maintaining each temperature for 60 min.

The $NH_3$ conversion was calculated using the following Eq. (1)

$$NH_3 \text{ conversion} = \frac{[NH_3]_{inlet} - [NH_3]_{outlet}}{\left(1 + [NH_3]_{outlet}\right) \times [NH_3]_{inlet}} \times 100\% \quad (1)$$

where $[NH_3]_{inlet}$ and $[NH_3]_{outlet}$ refer to the measured concentrations of $NH_3$ fed into and flowing out of the reactor.

The $H_2$ production rate, with the unit of mmol $H_2$ $g_{cat}^{-1}$ $min^{-1}$, was calculated from the $NH_3$ conversion ($X_{NH3}$) by the Eq. (2) below:

$$H_2 \text{ production rate} = \frac{WHSV \times X_{NH_3} \times 1.5}{V_m \times 60} \quad (2)$$

where WHSV is the weight hourly space velocity (30,000 mL $g_{cat}^{-1}$ $h^{-1}$), $X_{NH3}$ is the conversion of $NH_3$, and $V_m$ is the molar volume of gas at 25 °C and 1 atm (24 mL $mmol^{-1}$).

## Data availability

Source data are provided with this paper. The authors declare that the data supporting the findings of this study are available within the paper, its supplementary information files, and the Figshare repository (https://doi.org/10.6084/m9.figshare.24647937). Source data are provided with this paper.

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

## Acknowledgements
The authors thank Dr Anita D'ANGELO and the Powder Diffraction (PD) beamline at Australian Synchrotron, Melbourne, Australia. The authors thank the beamline BL01C at National Synchrotron Radiation Center, Hsinchu, Taiwan. This work was financially supported by the Hong Kong Polytechnic University BE3K (M.M.J.L.), PB1G (M.M.J.L.), ZVRP (Y.Z.), BE47 (M.Y.), ZE0C (M.Y.), ZE2F (M.Y.), and ZE2X (M.Y.); Innovation and Technology Fund (ITS/077/21) (M.M.J.L.); Research Grants Council of Hong Kong (General Research Fund No. 15307522) (Y.Z.); Shenzhen Municipal Science and Technology Innovation Commission, China (JCYJ20210324140811032) (M.M.J.L.); Department of Science and Technology of Guangdong Province, China (2021A1515010021) (M.M.J.L.). The TEM facility is funded by the Research Grants Council of Hong Kong (Project No. C5029-18E) (Y.Z.).

## Author contributions
M.M.J.L. supervised the project. M.M.J.L. and P.X. conceived the idea, designed synthesis approach, and performed catalyst synthesis, characterization, catalytic tests, and data analyses. Y.Z. and Z.H.X. carried out all the TEM and STEM characterizations and performed the related data analyses. T.S.W and Y.L.S. performed XAS measurements. M.Y. and T.Y. guided the high-throughput screening of solid-state using Materials Project. J.T.L. assisted with catalyst synthesis and methane dry-reforming reactions. G.C.L. and R.D.B. performed SXRD measurements, and G.C.L. performed the SXRD data analyses. P.X., Q.L., Y.Z., S.P.L., S.C.E.T., and M.M.J.L. wrote and revised the manuscript. All authors discussed the results and assisted with the manuscript preparation.

## Competing interests
The authors declare no competing interests.
