## [Peer Review File · Nature Communications]

Synthesis of core@shell metal@metal-oxide catalysts guided by Tammann temperatureREVIEWER COMMENTS

Reviewer #1 (Remarks to the Author):

The strong metal-support interaction (SMSI) plays a significant role in designing and modulating heterogeneous catalysts. This manuscript aims to provide a generalizable SMSI strategy guided by Tammann temperatures of materials to achieve functional oxide encapsulation of transition metal nanocatalysts. The results are interesting. The following concerns should be addressed before considering for publication.

1. For the Fig.1a, the authors depict the partial encapsulation of transition metal oxides, as far as I know, in the field of SMSI, the encapsulation of metal oxides by supports has never been reported. At the meantime, this manuscript is lack of relevant experimental evidences, such as TEM images, to demonstrate this schematic illustration.

2. Although the Tammann temperature can be applied as a descriptor to indicate the mobility of atoms in solid-state materials, it can not precisely predict the temperature for the SMSI construction as the formation of SMSI actually relates to many factors, such as particle size, facets of supports, etc. Will the Tammann temperature-guided strategy can be extended to other material systems as depicted in Fig1b? The authors need to provide more experimental results.

3. For the Fig.1c, the quality of EDS mapping data needs to be improved. For the encapsulation phenomenon achieved by SMSI, the Co NPs are actually surrounded by the covering overlayer, why there are almost no C and O signals in the area of Co NPs. Besides, the authors believe the shell or the encapsulation overlayer would be BaCO₃. However, it's not so convincing if the low-quality EDS data is the only evidence. In this manuscript, the authors provide the decomposition temperature for the BaCO₃, but the temperature may differ in the real material system with the existence of Co NPs. The authors should provide more convincing experimental evidences and mechanism explanations to elaborate this kind of SMSI.

4. For the functionalizing the BaCO₃ overlayer, the authors design this step during the Co NPs formation. Will things become different if the Al₂O₃ was added after the encapsulation of Co NPs? I think it will be a good complement for comparing the catalytic performance to the Co@ BaAl₂O₄.

5. For the Fig.2c, the corresponding HRTEM images and O signal images of the EDS mapping should be provided. Besides, why the C signals acquisition area is different from others?

Reviewer #2 (Remarks to the Author):

The manuscript describes an approach to metal core@shell/support systems leveraging thermal

properties (Tammann, reduction and decomposition temperatures). In this way the team is able to predict which materials fit into the temperature ranges that allow for the formation of the core@shell system. Thus, the approach is somewhat limited in terms of the materials that it may be utilized for. The resultant systems were then tested for the catalytic decomposition of NH₃ and dry reforming of methane. In both cases the results are pretty good. The real time observations of the systems forming are also very impressive.

There are numerous approaches to core@shell systems in the literature with some even demonstrating control of particle size and composition and others offering control of the pore size and thus size selective catalytic properties. Thus, the approach here is clever, interesting and novel but doesn't really provide the high impact that one expects from Nature communications.

Reviewer #3 (Remarks to the Author):

The authors present a generalized rational method based on the Tammann temperature, the decomposition temperature of the shell, and the reduction temperature of the core metal nanoparticle for synthesizing core@shell catalysts encapsulated by stable metal oxides. Initially, the authors conducted a thorough survey of the temperatures for various transition metals and metal oxides to identify potential candidates for stable core@shell structures. To validate this hypothesis, the authors prepared a core@shell catalyst using a combination of materials deemed most promising. Comprehensive characterization techniques, including in situ atmospheric STEM, in situ XRD, in situ Raman, and XAFS, were employed, offering deep insights into the encapsulation and synthesis mechanisms of the core@shell. This manuscript stands to serve as a valuable guide for researchers aiming to synthesize core@shell catalysts. However, adding illustrations detailing the synthesis strategy and providing more experimental data could further enhance the quality of this work, making it even more beneficial for the target audience in Nature Communications.

Comments are attached below:

1. The authors employed the T_{Tam} , T_{dec} , and T_{red} as key parameters in synthesis protocol, establishing that a temperature range of $T_{\text{Tam}} < T_{\text{red}} < T_{\text{dec}}$ is optimal for successful encapsulation. To substantiate this concept, the authors synthesized Co@BaAl₂O₄, with a Co nanoparticle core and a BaAl₂O₄ shell. However, the temperature attributes referenced were those of BaCO₃ rather than BaAl₂O₄. To more directly validate the proposition, it might be beneficial to attempt the synthesis of a Co@BaCO₃ structure. This result can also provide data to ignore any potential influence of the Al ion during synthesis.

2. To corroborate the concept presented in Fig. 1b, the authors synthesized additional core@shell structures, such as Cu@MgAl₂O₄ and Ni@BaAl₂O₄, demonstrating successful shell formation. How would the synthesis of Cu@CaCO₃ or Fe@MgCO₃? These combinations appear to deviate from the author's proposed criteria. It would be intriguing to ascertain if these exceptional combinations can be

successfully synthesized. If core@shell structures fail to form, it would further bolster the author's theory.

A point-by-point response to Reviewers' comments

We thank the editor and the three reviewers for their dedicated time and effort in reviewing our manuscript, as well as for their valuable comments and suggestions. We have carefully considered the feedback provided and have taken the necessary steps to enhance our manuscript, including the incorporation of additional analyses and discussions in this revised version.

In response to the comments received, we conducted additional experiments, which are summarized as follows:

(i) **Regulating of Co NP sizes in our developed methodology:** We have adjusted the Co NP sizes in our developed methodology, leading to the successful synthesis of Co@BaAl₂O₄ core@shell structures with varying Co NP sizes.

(ii) **Control experiments to verify the synthesis rationale of our developed methodology:** To validate the synthesis rationale, we have introduced control examples by subjecting precursor mixtures of Fe₂O₃ + BaCO₃ + Al₂O₃ and CuO + BaCO₃ + Al₂O₃ to H₂ thermal treatment. The severe sintering of Fe and Cu nanoparticles is observed, affirming the validity of the thermal encapsulation strategy guided by Tammann temperature.

(iii) **Improved identification of Co@BaCO₃ core@shell structure:** To provide a more comprehensive microscopic characterization of the Co@BaCO₃ core@shell structure, we have included supplementary EELS characterization with comprehensive element mapping.

(iv) **Studying the effect of Co NPs presence on BaCO₃ decomposition:** We have conducted an investigation into the influence of Co NPs on the decomposition of BaCO₃ through temperature-programmed decomposition of Co₃O₄ and BaCO₃ mixtures in an H₂ atmosphere. The results indicate that the presence of Co NPs (reduced from Co₃O₄) does not affect the decomposition process of BaCO₃.

We have provided a point-by-point response to the reviewers' comments, and the corresponding changes in our revised main manuscript and Supplementary Materials (SI) have been highlighted in yellow. In addition, we have implemented several additional minor corrections in both the main text and the SI to enhance the clarity and accuracy of the manuscript. These modifications have also been highlighted in yellow for your convenience and easy reference.

Response to Reviewer #1

The strong metal-support interaction (SMSI) plays a significant role in designing and modulating heterogeneous catalysts. This manuscript aims to provide a generalizable SMSI strategy guided by Tammann temperatures of materials to achieve functional oxide encapsulation of transition metal nanocatalysts. The results are interesting. The following concerns should be addressed before considering for publication.

Response: We sincerely appreciate the reviewer for providing positive feedback on our work and offering valuable suggestions. The reviewer's comments have significantly enriched the content and insights presented in our paper. We have carefully addressed each comment individually, as outlined below.

1. For the Fig.1a, the authors depict the partial encapsulation of transition metal oxides, as far as I know, in the field of SMSI, the encapsulation of metal oxides by supports has never been reported. At the meantime, this manuscript is lack of relevant experimental evidences, such as TEM images, to demonstrate this schematic illustration.

Response: We appreciate the Reviewer for raising this critical point. We apologize for any confusion caused by the previous version's potentially misleading schematic illustration, which did not clearly convey our synthesis strategy.

In the previous version, Fig. 1a's schematic of "support diffusion" may have given the impression that a "partial encapsulation" process occurs when the support reaches its Tammann temperature. However, our intention was not to demonstrate or imply such a phenomenon, as no such observation has been reported in the literature due to the greater stability of metal oxides compared to metal nanoparticles. From our *in-situ* STEM experiment (Fig. 3b, Supplementary Fig. 22b and Supplementary Movie I), we observed the diffusion of the support and its contact with $M_T O_x$ at T_{Tam} . The enhanced diffusion of the support and its intimate contact with the $M_T O_x$ at this stage validate the main concept of our strategy, which is to have a mobile support material. Following the "support diffusion" process, as the temperature increases further to $T_{red}(M_T O_x)$, the $M_T O_x$ undergoes reduction to form metal nanoparticles (M_T NPs) with higher surface energy. To minimize the surface energy of the M_T NPs, the support tends to migrate to the metal surface and form an overlayer, resulting in the achieved encapsulation structure known as strong metal-support interactions (SMSI). This SMSI process is clearly demonstrated through STEM imaging in the $Co@BaCO_3$ sample at $T_{red}(Co_3O_4)$ in the H_2 atmosphere (Fig. 1c, and Supplementary Fig. 8).

To highlight the distinction between the "support diffusion at $T_{Tam}(\text{support})$ " and "SMSI encapsulation at $T_{red}(M_T O_x)$ " stages, we have revised Fig. 1a in the manuscript, which is displayed as **Fig. R1** below, and provided further clarification of the "support diffusion" step on **Page 2, Line 85** in the revised manuscript:

"According to this rule, the support can facilitate more efficient diffusion, acting as a "mobile media" to establish contact with the $M_T O_x$ at $T_{Tam}(\text{support})$. During the formation of M_T NP at $T_{red}(M_T O_x)$, encapsulation can take place due to the tendency to reduce the surface energy of M_T NPs. This process helps prevent the excessive growth of M_T particles."

Figure R1 (as Figure 1a of the revised manuscript). Schematic illustration of the designed synthesis rationale for this thermal-induced encapsulation strategy guided by Tammann temperature.

2. Although the Tammann temperature can be applied as a descriptor to indicate the mobility of atoms in solid-state materials, it can not precisely predict the temperature for the SMSI construction as the formation of SMSI actually relates to many factors, such as particle size, facets of supports, etc. Will the Tammann temperature-guided strategy can be extended to other material systems as depicted in Fig1b? The authors need to provide more experimental results.

Response:

We highly appreciate the valuable comments provided by the reviewer. It is true that the empirical Tammann temperature of the support cannot precisely predict the temperature for constructing SMSI. Its main purpose is to guide the selection of suitable materials that meet the first criterion of SMSI, which is having a mobile support. In our *in-situ* experimental observations, the construction of SMSI occurs at the temperature when the $M_T O_x$ is reduced to M_T NPs at $T_{red}(M_T O_x)$. The further modification of the overlayer is then governed by the formation temperature of the targeted solid-state reaction product, such as the conversion of $BaCO_3$ and Al_2O_3 to $BaAl_2O_4$ in our study.

We agree with the reviewer that the overall synthesis strategy can be influenced by various factors, which in turn can affect the construction of SMSI. This is due to the involvement of multiple synthesis parameters and multi-component precursors in the proposed one-step thermal treatment. While this is a reasonable concern, we also see significant potential in controlling and regulating the SMSI core@shell structure by manipulating these synthesis parameters. This provides opportunities to diversify the SMSI configurations.

In light of this, we have conducted additional experiments and investigations to examine the effect of particle size, as suggested by the reviewer, on the constructed SMSI structure. To control the particle sizes of the precursor mixtures, we introduced an air calcination step before the H_2 thermal treatment. As shown in the XRD size analysis results (**Fig. R2a** and **Table R1**), we synthesized mixture precursors with different Co_3O_4 sizes by adjusting the air-calcination temperature. After thermal treatment with H_2 , the resulting core Co NPs' crystallite sizes increased in the samples experienced higher air calcination temperatures, as demonstrated in the TEM and XRD results (**Figs. R2b-e** and **Table R1**). This outcome indicates that employing different particle sizes indeed leads to different SMSI core@shell configurations. With this in mind, we are motivated to explore other factors and create versatile SMSI configurations in our future research.

We have included the aforementioned additional experimental result in the revised SI (Pages 28-29):

Figure R2 (as Supplementary Figure 18 of the revised SI). Effect of Co₃O₄ precursor particle sizes on SMSI formation. (a) XRD patterns of Co₃O₄ + BaCO₃ + Al₂O₃ mixture precursors with different air-calcination temperatures. (b) XRD patterns of final obtained Co@BaAl₂O₄ core@shell structure with different air-calcination temperatures. (c-e) TEM images and Co NPs size distribution measured from low-magnitude TEM images of final obtained Co@BaAl₂O₄ core@shell structure with different air-calcination temperatures: (c) without calcination, (d) 500°C calcined, and (e) 700°C calcined.

Table R1 (as Supplementary Table 10 of the revised SI). Estimation of Co_3O_4 crystallite sizes in C(temp.)-40Co:20Ba:40Al and the associated Co NPs crystallite sizes in C(temp.)-R(700)-40Co:20Ba:40Al.

Sample	Phase (Peak selected for crystallite size calculation)	Crystallite size calculated using the Debye-Scherrer equation		Co NPs size distribution from TEM, nm	BaAl ₂ O ₄ overlayer thickness from TEM, nm
		FWHM, °	Crystallite size, nm		
40Co:20Ba:40Al	No obvious Co_3O_4 (311) peak	/	/	/	/
C(500)-40Co:20Ba:40Al	Co_3O_4 (311)	0.702	12.0	/	/
C(700)-40Co:20Ba:40Al	Co_3O_4 (311)	0.628	13.4	/	/
R(700)-40Co:20Ba:40Al	Co (111)	0.927	9.2	6 ~ 12	c.a.4.0
C(500)-R(700)-40Co:20Ba:40Al	Co (111)	0.613	14.7	10 ~ 20	c.a. 4.1
C(700)-R(700)-40Co:20Ba:40Al	Co (111)	0.371	23.2	15 ~ 25	c.a. 3.5

Additionally, the reviewer raised a question regarding the applicability of the Tammann temperature-guided strategy to other material systems depicted in Fig. 1b. We want to clarify that our original submission already includes examples of other successful SMSI core@shell combinations (**Page 9, Line 307** in the revised manuscript). To provide convenience for the reviewer, we have highlighted the relevant discussions below:

"A core@shell catalyst design and synthesis platform with broad applicability

We have successfully demonstrated that low- T_{Tam} compounds can facilitate thermal-induced SMSI core@shell construction, and subsequent solid-state reactions for overlayer functionalization enable thermal catalytic applicability. Furthermore, to showcase the general applicability of our proposed approach, we also test other M_{T} and $M_{\text{AEC}}\text{CO}_3$ combinations in the encapsulable window (shadowed in blue in Fig. 1b). The results have validated the effectiveness of our thermal encapsulation and sintering prevention strategy guided by Tammann temperature (Fig. 4e which is displayed as **Fig. R3** below, Supplementary Fig. 32 and 33, and Supplementary Table 20). More convincingly, the replacement of Al_2O_3 with TiO_2 for the overlayer solid-state reaction enables us to achieve BaTiO_3 encapsulation on Co NPs (Fig. 4e which is displayed as **Fig. R3** below, Supplementary Fig. 34, and Supplementary Table 20), demonstrating the broad applicability of our strategy across various material matrices (Supplementary Fig. 35)."

Figure R3 (as **Figure 4e** of the revised manuscript). STEM images and EDS element distribution maps of Cu@MgAl₂O₄, Ni@BaAl₂O₄, and Co@BaTiO₃.

3. For the Fig.1c, the quality of EDS mapping data needs to be improved. For the encapsulation phenomenon achieved by SMSI, the Co NPs are actually surrounded by the covering overlayer, why there are almost no C and O signals in the area of Co NPs. Besides, the authors believe the shell or the encapsulation overlayer would be BaCO₃. However, it's not so convincing if the low-quality EDS data is the only evidence. In this manuscript, the authors provide the decomposition temperature for the BaCO₃, but the temperature may differ in the real material system with the existence of Co NPs. The authors should provide more convincing experimental evidences and mechanism explanations to elaborate this kind of SMSI.

Response:

We appreciate the reviewer for raising this concern and providing suggestions regarding our presentation of the Co@BaCO₃ results. We apologize for not clarifying that the mapping technique used was electron energy loss spectroscopy (EELS) in our original submission. In response, we have clearly indicated that the mapping was performed using EELS in the revised Fig. 1c caption within the manuscript (**Page 3, Line 117**).

The limited visibility of the C and O signals in the EELS spectra of the overlayer can be attributed to the thin shell thickness of the achieved SMSI structure. To provide more convincing evidence, we have included another set of EELS mapping data for the SMSI Co@BaCO₃ with a slightly thicker shell in the revised SI on **Page 14** (Supplementary Fig. 8c), which is displayed in **Fig. R4c** below. The color-coded maps clearly illustrate the distribution of Ba, O, and C elements in the overlayer surrounding the core Co particle. To further validate the presence of C and O elements in the covering overlayer of the core Co NPs, we extracted EELS spectra from the marked locations (**Fig. R4c**) in both the core and the shell, as shown in **Fig. R4d-e**. These extracted spectra, including the O K-edge and C K-edge spectra, demonstrate the presence of both O and C elements in both the core covering overlayer and the shell regions. The observed O and C elements are from the BaCO₃ structure, as the XRD data in **Fig. R4a** indicate the presence of BaCO₃ and *fcc* Co as the only observed compounds in this sample. Additionally, as depicted in **Fig. R4f** for the Ba L-edge and Co L-edge, it is evident that the core exhibits a high-intensity Co signal with a weak Ba signal represented by a small hump observed beside

the Co L₂-edge, as indicated by the arrow. These findings collectively confirm the presence of a BaCO₃ overlayer encapsulating the Co NPs.

Figure R4 (as Supplementary Figure 8 of the revised SI). Proof-of-concept demonstration by constructing Co@BaCO₃ encapsulation configuration. (a) XRD pattern of C(500)-R(500)-67Co:33Ba. (b) Co NP size distribution measured from low magnification TEM images of C(500)-R(500)-67Co:33Ba. (c) EELS element distribution maps of C(500)-R(500)-67Co:33Ba, including Co, Ba, O and C elements. (d-f) The EELS spectra of O-K, C-K, and Co-L, Ba-L edges extracted from the core and the overlayer.

On the other hand, the reviewer's concern regarding potential temperature variations in the used material systems due to the presence of Co NPs has been duly noted, and we appreciate this feedback. To thoroughly investigate the impact of Co NPs on the decomposition temperature of BaCO₃, detail of which is on **Page 10** of the revised SI, we conducted experiments involving the mixture of BaCO₃ and Co₃O₄ prepared by co-precipitation (confirmed by XRD in Supplementary Fig. 6f, upper panel). These precursor mixtures were subjected to H₂/Ar treatment, with mass spectrometry (MS) tracking of H₂ (m/z = 2), H₂O (m/z = 18), and CO (m/z = 28). **Fig. R5b** illustrates the MS signals as a function of temperature during the H₂ thermal treatment process.

From the data, it is evident that Co₃O₄ undergoes reduction to form *fcc* Co NPs within the temperature range of 300 ~ 500 °C (XRD analysis confirming the presence of BaCO₃ and the formed *fcc* Co in

Supplementary Fig. 6f, lower panel). During this reduction process, H₂O is released, and no CO is detected in this temperature range. This observation validates that BaCO₃ remains undecomposed when Co NPs are formed through the reduction of Co₃O₄. As the temperature further increases, the CO peak becomes evident around 700 °C, peaking at 909 °C, signifying the decomposition of BaCO₃. This result aligns with the decomposition behavior of pure BaCO₃ (which peaked at 916 °C), as presented in the original submission (Fig. R5a). Thus, these findings confirm that the presence of Co NPs has no noticeable effect on the decomposition process of BaCO₃.

Figure R5 (as Supplementary Figure 6d-e of the revised SI). Determination of the decomposition temperature (T_{dec}) of BaCO₃ with and without Co₃O₄. (a) MS signals during the H₂ temperature-programmed decomposition of pure BaCO₃. (b) MS signals during the H₂ temperature-programmed decomposition of BaCO₃ + Co₃O₄ mixture precursor.

4. For the functionalizing the BaCO₃ overlayer, the authors design this step during the Co NPs formation. Will things become different if the Al₂O₃ was added after the encapsulation of Co NPs? I think it will be a good complement for comparing the catalytic performance to the Co@BaAl₂O₄.

Response:

We appreciate the reviewer's suggestion, and we acknowledge the valid point raised regarding the potential variation in the catalytic performance of Co@BaAl₂O₄ when Al₂O₃ is added after the BaCO₃ encapsulation of Co NPs. It is true that many synthesis parameters can influence the final configuration of SMSI core@shell products, subsequently impacting the resulting catalyst structures. However, it is important to note that our primary focus in the current article was on presenting our novel SMSI synthesis strategy, elucidating the underlying mechanisms, and showcasing the applicability of the fabricated SMSI catalysts in thermal catalytic applications. Consequently, performing additional experiments to explore the relationship between synthesis parameters, catalyst structure, and performance would expand beyond the scope of our present study.

Nevertheless, we fully appreciate that the reviewer's suggestion is both useful and reasonable. Such an approach could provide valuable insights into enhancing the performance of the proposed SMSI core@shell catalyst. With this in mind, we have initiated careful experiments to investigate the correlations between synthesis, structure, and performance for Co@BaAl₂O₄ and related material systems. We intend to share the results of these experiments in our future research outcomes. We sincerely apologize for not including the suggested experiment in this current revision, and we hope the reviewer can accept our apology.

5. For the Fig.2c, the corresponding HRTEM images and O signal images of the EDS mapping should be provided. Besides, why the C signals acquisition area is different from others?

Response:

We thank the reviewer for this comment. In the revised manuscript, we have included the HRTEM image and the O signal image from EDS mapping in Supplementary Fig. 11d (on **Page 18** of the revised SI), now in this letter presented as **Fig. R6**. The HRTEM image clearly demonstrates the core@shell structure, with Ba, Al, and O elements surrounding the Co element.

Fig. R6 (as Supplementary Figure 11d of the revised SI). O signal images of the EDS mapping. Atomic-scale STEM images and EDS element distribution maps (including Co, Ba, Al, and O elements) of Co@BaAl₂O₄.

In addition, we apologize for any confusion caused by the misleading figure label in Fig. 2c. The figure at the upper left corner has been corrected to accurately represent the HRTEM images of Co@BaAl₂O₄. To address this issue and prevent any further confusion, we have adjusted the figure label "c" to a more appropriate position. The revised Fig 2c is attached below (**Fig. R7**).

Figure R7 (as Figure 2c of the revised manuscript). STEM image and EDS element distribution maps of Co@BaAl₂O₄.

Response to Reviewer #2

The manuscript describes an approach to metal core@shell/support systems leveraging thermal properties (Tammann, reduction and decomposition temperatures). In this way the team is able to predict which materials fit into the temperature ranges that allow for the formation of the core@shell system. Thus, the approach is somewhat limited in terms of the materials that it may be utilized for.

The resultant systems were then tested for the catalytic decomposition of NH₃ and dry reforming of methane. In both cases the results are pretty good. The real time observations of the systems forming are also very impressive. There are numerous approaches to core@shell systems in the literature with some even demonstrating control of particle size and composition and others offering control of the pore size and thus size selective catalytic properties. Thus, the approach here is clever, interesting and novel but doesn't really provide the high impact that one expects from Nature communications.

Response:

We genuinely appreciate the positive comments provided by the reviewer regarding our synthesis and characterization approaches, as well as the promising performance of our catalytic applications. Your feedback is valuable to us.

Firstly, we would like to acknowledge the reviewer's comment on the abundance of existing works in core-shell synthesis in the literature. While it is true that this field has seen significant research, we believe our approach stands out due to the unique and innovative use of the Tammann temperature guidance for achieving a generalizable encapsulation and modification.

Regarding the reviewer's concern about limited material options in our proposed selection rule, we would like to emphasize that there are still numerous low-Tammann temperature materials available. Examples include chloride, sulfate, nitrate, and others, which can effectively be utilized to construct size-controlled encapsulation structures for various transition metal nanoparticles.

Furthermore, our Tammann temperature-guided core@shell synthesis method offers a wide array of material synthesis parameters to explore. These include composition control (as validated in Fig. 4e on **Page 8** of the revised Manuscript), size control (as presented in Supplementary Fig. 18 on **Page 28** of the revised SI), *in-situ* metal oxide reduction or the use of pre-made metal nanoparticles, the timing of adding overlay modifiers, and many more. These factors provide ample room for further optimisation and advancement in future studies.

Additionally, we envision that as we continue to validate diverse core@shell structures through experimental exploration, computational calculations can be integrated to facilitate rapid material selection from a vast matrix for overlayer solid-state reactions. This integration has the potential to unlock a wide range of possibilities for catalyst development.

To summarise, our work represents the beginning of an innovative and evolving development, with continuous growth opportunities. The current study has offered a valid and impactful approach for preparing active, durable, and potentially versatile core@shell catalysts. With continuous efforts and exploitations, the Tammann temperature-guided SMSI construction approach can significantly advance performance in various applications. We firmly believe that our work has growing significance and we can see its potential contributions to the diverse and active catalysis community. Therefore, we sincerely request the reviewer and editor to consider publishing this work in Nature Communications. Thank you.

Response to Reviewer #3

The authors present a generalized rational method based on the Tammann temperature, the decomposition temperature of the shell, and the reduction temperature of the core metal nanoparticle for synthesizing core@shell catalysts encapsulated by stable metal oxides. Initially, the authors conducted a thorough survey of the temperatures for various transition metals and metal oxides to identify potential candidates for stable core@shell structures. To validate this hypothesis, the authors prepared a core@shell catalyst using a combination of materials deemed most promising. Comprehensive characterization techniques, including in situ atmospheric STEM, in situ XRD, in situ Raman, and XAFS, were employed, offering deep insights into the encapsulation and synthesis mechanisms of the core@shell. This manuscript stands to serve as a valuable guide for researchers aiming to synthesize core@shell catalysts. However, adding illustrations detailing the synthesis strategy and providing more experimental data could further enhance the quality of this work, making it even more beneficial for the target audience in Nature Communications.

Comments are attached below:

Response:

We express our gratitude for the reviewer's review of our manuscript and for providing positive and constructive feedback. The reviewer's comments have greatly contributed to enhancing the depth and clarity of our paper. We have added illustrations detailing the synthesis strategy in the revised SI Supplementary Fig. 35 (Page 49 of the revised SI), including all the successful core@shell examples and control examples failing to form the core@shell structures.

We have also thoroughly addressed each of the reviewer's comments, as outlined below:

1. The authors employed the T_{Tam} , T_{dec} , and T_{red} as key parameters in synthesis protocol, establishing that a temperature range of $T_{\text{Tam}} < T_{\text{red}} < T_{\text{dec}}$ is optimal for successful encapsulation. To substantiate this concept, the authors synthesized Co@BaAl₂O₄, with a Co nanoparticle core and a BaAl₂O₄ shell. However, the temperature attributes referenced were those of BaCO₃ rather than BaAl₂O₄. To more directly validate the proposition, it might be beneficial to attempt the synthesis of a Co@BaCO₃ structure. This result can also provide data to ignore any potential influence of the Al ion during synthesis.

Response:

We appreciate the reviewer for highlighting the significance of the Co@BaCO₃ structure in validating our proposition. It is important to clarify that our original submission already incorporated this attempt, and we successfully obtained the Co@BaCO₃ structure through the proposed SMSI strategy. To facilitate the reviewer's assessment, we have summarised the relevant discussions below:

Uniformly distributed Co NPs encapsulated in BaCO₃ (i.e., Co@BaCO₃) is obtained through one-step thermal treatment at $T_{\text{red}}(\text{Co}_3\text{O}_4)$ in H₂ atmosphere. As shown in Fig. 1c (now in this letter presented as Fig. R8) and Supplementary Fig. 8c (now in this letter presented as Fig. R9), the STEM and elemental EELS mapping reveal that the synthesized Co@BaCO₃ exhibits a core@shell structure, with a core enriched with Co element, and an overlayer enriched with Ba, C, and O elements. The *fcc* Co and BaCO₃ phases are further confirmed by the XRD results of the final products (Supplementary Fig.

8a). These results demonstrate the viability of our proposed strategy for the convenient thermal encapsulation on the low melting point M_T NPs with encapsulating materials guided by Tammann temperatures.

Figure R8 (as **Figure 1c** of the revised manuscript). TEM, STEM images and EELS element distribution maps of Co@BaCO₃.

Figure R9 (as **Supplementary Figure 8c** of the revised SI). Additional set of STEM and EELS element distribution maps of Co@BaCO₃, including Co, Ba, O and C elements.

2. To corroborate the concept presented in Fig. 1b, the authors synthesized additional core@shell structures, such as Cu@MgAl₂O₄ and Ni@BaAl₂O₄, demonstrating successful shell formation. How would the synthesis of Cu@CaCO₃ or Fe@MgCO₃? These combinations appear to deviate from the author's proposed criteria. It would be intriguing to ascertain if these exceptional combinations can be successfully synthesized. If core@shell structures fail to form, it would further bolster the author's theory.

Response:

We greatly appreciate the reviewer's insightful comment and suggestion. In response to this, we conducted experiments involving M_TO_x-M_{AE}CO₃ combinations where the $T_{\text{red}}(M_{\text{T}}O_x)$ falls either outside or on the edge of the encapsulable window. The results are added in Supplementary Fig. 16 and 17 (**Pages 26-27** of the revised SI, now in this letter presented as **Figs. R10 and R11**). Specifically, we examined the Fe₂O₃-BaCO₃ and CuO-BaCO₃ combinations. In both cases, after thermally treating mixtures of 40Fe:20Ba:40Al and 40Cu:20Ba:40Al in a H₂ atmosphere, we observed no encapsulation, and the resulting Fe and Cu NPs displayed severe coalescence and broad size distributions (see **Figs. R10a-d and R11a-d** below).

For the Fe₂O₃ + BaCO₃, it's important to note that BaCO₃ decomposes into high- T_{Tam} BaO before Fe NPs can form from Fe₂O₃. Consequently, Fe NPs undergo substantial sintering without the protective encapsulation effect provided by BaCO₃ (**Fig. R10**). In the case of CuO + BaCO₃, on the other hand, when the temperature reached the T_{red} of CuO (~250 °C), which is still lower than the T_{Tam} of BaCO₃ (269 °C), the limited mobility of BaCO₃ is insufficient to form an encapsulation layer on Cu NPs. This also results in significant sintering of Cu NPs (**Fig. R11**).

These control experiments serve as valuable cross-check examples, affirming the validity of our synthesis rationale for the new thermal encapsulation strategy guided by Tammann temperature.

Figure R10 (as Supplementary Figure 16 of the revised SI). Control sample which falls either outside or on the edge of the encapsulable window: Fe₂O₃ + BaCO₃ + Al₂O₃. (a) XRD patterns of C(500)-40Fe:20Ba:40Al precursor and final obtained C(500)-R(700)-40Fe:20Ba:40Al. (b-c) TEM images of final obtained C(500)-R(700)-40Fe:20Ba:40Al. (d) Fe NP size distribution measured from low-magnification TEM images. (e) Schematic illustration of the reasons for the failure of the Fe@BaAl₂O₄ core@shell structure synthesis from Fe₂O₃ + BaCO₃ + Al₂O₃ mixture precursors.

Figure R11 (as Supplementary Figure 17 of the revised SI). Control sample which falls either outside or on the edge of the encapsulable window: $\text{CuO} + \text{BaCO}_3 + \text{Al}_2\text{O}_3$. (a) XRD patterns of C(500)-40Cu:20Ba:40Al precursor and final obtained C(500)-R(700)-40Cu:20Ba:40Al. (b-c) TEM images of final obtained C(500)-R(700)-40Cu:20Ba:40Al. (d) Cu NP size distribution measured from low-magnitude TEM images. (e) Schematic illustration of the reasons for the failure of the $\text{Cu}@ \text{BaAl}_2\text{O}_4$ core@shell structure synthesis from $\text{CuO} + \text{BaCO}_3 + \text{Al}_2\text{O}_3$ mixture precursors.

REVIEWERS' COMMENTS

Reviewer #1 (Remarks to the Author):

The comments were appropriately addressed by the authors, and the manuscript quality was improved as well. I recommend this manuscript for publication in its current form.

Reviewer #3 (Remarks to the Author):

Upon thorough re-evaluation of the revised manuscript, I am pleased to recommend it for acceptance. The authors have commendably addressed the concerns raised during the initial review process by performing additional experiments that not only strengthen the core arguments of their study but also substantially contribute to the scientific rigor of their work.

1) The authors have effectively regulated the sizes of Co nanoparticles (Co NPs), leading to the successful synthesis of Co@BaAl₂O₄ core@shell structures with precise Co NP dimensions. This demonstrates a significant advancement in their proposed methodology and adds considerable value to the field of materials synthesis.

2) Control experiments, introduced to validate the synthesis rationale, have provided clear evidence supporting the thermal encapsulation strategy, which is further substantiated by the observed sintering behaviors of Fe and Cu nanoparticles under similar conditions. This comparison with Fe₂O₃ and CuO adds a layer of validation to their innovative approach.

3) Moreover, the enhanced characterization of the Co@BaCO₃ core@shell structure through supplementary EELS characterization and comprehensive element mapping offers a more in-depth understanding of the structural and compositional subtleties of the synthesized materials.

A point-by-point response to Reviewers' comments

We would like to sincerely express our gratitude to all the Reviewers for their valuable and constructive comments, which have greatly contributed to improving the quality of our manuscript. We would also like to extend our heartfelt appreciation to the Senior Editor, Eric J. Piechota, for his professionalism and dedication in overseeing the review process. Furthermore, we would like to thank the entire technological editorial team for their time and effort in preparing the comprehensive checklist. In the subsequent section, we have addressed the specific points raised by Reviewer #1 and Reviewer #3.

Response to Reviewer #1

The comments were appropriately addressed by the authors, and the manuscript quality was improved as well. I recommend this manuscript for publication in its current form.

Response: Thank you so much for your time and expertise in evaluating our work. We are glad to hear that the comments have been effectively addressed. Your recommendation for publication is encouraging and deeply appreciated.

Response to Reviewer #3

Upon thorough re-evaluation of the revised manuscript, I am pleased to recommend it for acceptance. The authors have commendably addressed the concerns raised during the initial review process by performing additional experiments that not only strengthen the core arguments of their study but also substantially contribute to the scientific rigor of their work.

Response: Thank you for your meticulous re-evaluation and positive recommendation for accepting the revised manuscript. We greatly appreciate your recognition of our efforts and the confirmation that the revised version has met your expectations. Our primary objective is to make a meaningful contribution to the scientific community, and we firmly believe that the incorporation of additional experiments has successfully advanced this goal. We are sincerely grateful for your support and confidence in our work.

1) The authors have effectively regulated the sizes of Co nanoparticles (Co NPs), leading to the successful synthesis of Co@BaAl₂O₄ core@shell structures with precise Co NP dimensions. This demonstrates a significant advancement in their proposed methodology and adds considerable value to the field of materials synthesis.

Response: Thank you so much for your insightful comment. We are glad that you recognize

the effectiveness of our approach in regulating the sizes of Co nanoparticles (Co NPs) and successfully synthesizing Co@BaAl₂O₄ core@shell structures with precise Co NP dimensions. We appreciate your acknowledgement of the significant advancement our methodology represents and the value it adds to the field of materials synthesis. Your feedback further motivates us to continue our research and contribute more to this field.

2) Control experiments, introduced to validate the synthesis rationale, have provided clear evidence supporting the thermal encapsulation strategy, which is further substantiated by the observed sintering behaviors of Fe and Cu nanoparticles under similar conditions. This comparison with Fe₂O₃ and CuO adds a layer of validation to their innovative approach.

Response: We sincerely appreciate your perceptive comment and valuable insights regarding the control experiments we performed to validate the synthesis rationale and confirm the effectiveness of the thermal encapsulation strategy. We are delighted that our research has presented clear evidence and earned your appreciation. Your valuable feedback serves as motivation for us to continue exploring this novel strategy and its impacts.

3) Moreover, the enhanced characterization of the Co@BaCO₃ core@shell structure through supplementary EELS characterization and comprehensive element mapping offers a more in-depth understanding of the structural and compositional subtleties of the synthesized materials.

Response: Thank you for your valuable comment. We greatly appreciate your recognition of the enhanced characterization techniques we utilized. These techniques have indeed yielded a more profound understanding of the structural and compositional intricacies of the synthesized materials, which truly improves the quality of this work.